# High-Power Ultrasound in Gas Phase: Effects on the Bioactive Compounds Release from Red Bell Pepper during In Vitro Gastrointestinal Digestion

**DOI:** 10.3390/antiox12020356

**Published:** 2023-02-02

**Authors:** Cristina Reche, Carmen Rosselló, Valeria Eim, Alberto Edel León, Susana Simal

**Affiliations:** 1Department of Chemistry, University of the Balearic Islands, Ctra. Valldemossa km. 7.5, 07122 Palma de Mallorca, Spain; 2Facultad de Ciencias Agropecuarias, Universidad Nacional de Córdoba, ICYTAC (CONICET-UNC), CC 509, Córdoba 5000, Argentina

**Keywords:** ultrasound, gastrointestinal digestion, antioxidant activity, phenolic compounds, microstructure, image analysis

## Abstract

High-power ultrasound in gas-phase (28.8 kW/m^3^ for 120 min at 17.5 ± 0.3 °C) has been evaluated as a pre-treatment to enhance the release of antioxidants and phenolic compounds from red bell pepper during digestion. The moisture content decreased (34 ± 4%) while both the antioxidant activity (between 4 ± 1% and 21 ± 1%) and the phenolic compounds content (37 ± 4%) increased after the treatment. Moreover, microstructural changes were observed in the treated sample, with the appearance of breaks in the plant tissue, cell shrinkage, and an increased number of cells per area unit (28 ± 2%). Bioaccessibility was determined by in vitro gastrointestinal digestion. The total release of antioxidants and phenolic compounds after gastrointestinal digestion was 22–55% higher and 45 ± 7% higher, respectively, in the sonicated sample, with cell swelling and a 9.2 ± 0.1% higher number of cells per area unit. Therefore, the ultrasound treatment caused microstructural changes in the red bell pepper tissue, which could help to explain the higher release of bioactive compounds.

## 1. Introduction

High-power ultrasound (HPU) has become a recurrent treatment technology in processes such as drying [1], extraction [2], osmotic dehydration [3], enzymatic hydrolysis [4], etc., for process intensification. Ultrasound are mechanical waves that require an elastic medium to propagate through it [5]. Therefore, air would be a suitable medium for its propagation; however, its use as a propagation medium has mainly been used in studies involving the drying process [6].

One of the main effects of high-power ultrasound traveling through a solid is the “sponge effect”, which consists of alternating contractions and expansions of the solid. This effect can generate modifications of the structure, such as the creation of microscopic channels or the rupture of the cell walls [7,8]. 

The knowledge of microstructural changes due to processing is essential for understanding and predicting changes in the physicochemical properties of foods and could provide more information and a better understanding of the relationships in the structure and function of foods [9]. Several studies have qualitatively analyzed the microstructure of plant matrices after ultrasound application in the process of drying, extraction, freezing, etc. [10,11,12]. However, the characterization and the quantification of the microstructural changes are relevant for different applications, and only a few studies have quantified the effects by image analysis using computer technology. 

During digestion, the bioactive compounds have to be released from the food matrix (bioaccessibility) before they can be absorbed into the gastrointestinal tract (bioavailability). Therefore, it is not only interesting to know the number of bioactive compounds present in the original plant matrix but also the total bioaccessible amount [13]. In vitro digestion simulates the physiological conditions of in vivo digestion and is a useful tool to study and understand nutrient changes, interactions, and bioaccessibility. Different studies have demonstrated that the release and absorption of constituent nutrients are influenced by both the processing conditions and the cellular structure of the food [14,15]. González et al. [16] studied the effect of hot air drying on tannins and oxidative activity by in vitro gastrointestinal digestion of “Rojo Brillante” persimmon. They observed that hot air-dried persimmon exhibited the highest rate of soluble tannin recovery. Dalmau et al. [17] studied the effect of freeze-drying on beetroot and observed that samples after drying showed a higher release of phenolic compounds and antioxidant activity during in vitro digestion than the raw sample.

Bell pepper (*Capsicum annuum* L) pericarp accumulates considerable amounts of a wide range of phytochemicals, being considered one of the vegetables with the highest vitamin C content. It also contains vitamins A and E, fiber, as well as bioactive compounds such as phenolic compounds, carotenoids, and antioxidants with known properties that confer protection against carcinogenic compounds and delay the aging process, making red bell pepper an important healthy food [18,19]. Nevertheless, the absorption and metabolism of phenolic and antioxidant compounds in the digestive tract determine their biological properties [20]. 

To the best of our knowledge, the release of antioxidants and phenolic compounds after the application of high-power ultrasound (HPU) in the gas phase has not been studied, neither its effect on the microstructure by image analysis. The main objective of this work was to evaluate if the application of HPU in gas phase for 120 min could help to intensify the release of bioactive compounds (antioxidants and phenolic compounds) from red bell pepper and to quantify the microstructure changes by SEM and optical microscopy to justify the increased release of bioactive compounds when ultrasound pre-treatment is applied.

## 2. Materials and Methods

### 2.1. Chemicals

Ethanol 96% (*v*/*v*) extra pure, sodium bicarbonate (NaHCO_3_), magnesium chloride hexahydrate (MgCl_2_·6H_2_O), ammonium carbonate ((NH_4_)_2_CO_3_), and hydrochloric acid (HCl) 37% (*v*/*v*) were purchased from Scharlau (Barcelona, Spain). Potassium chloride (KCl), potassium phosphate monobasic (KH_2_PO_4_), sodium chloride (NaCl), calcium chloride dihydrate (CaCl_2_·2H_2_O), and sodium hydroxide (NaOH) 1M were purchased from Panreac (Barcelona, Spain). Alpha-amylase used for in vitro simulated gastrointestinal digestion was purchased from MP Biomedicals (Aurora, Ohio, USA), and pepsin from porcine gastric mucosa, pancreatin from porcine pancreas (8USP), and bile bovine were purchased from Sigma Aldrich (Steinheim, Germany). 

### 2.2. Sample Preparation

Red bell pepper (*Capsicum annuum* L) grown in Spain, was purchased from a local supermarket and stored at 4 °C for a maximum of one week. The pedicel, placenta, seeds, and apex were removed from the red bell pepper and the pericarp was cut into 18 mm diameter discs. Discs with a 6 ± 1 mm thickness were selected and processed immediately after cutting.

The set-up used in this study was previously described by Vallespir et al. [21]. The red bell pepper discs were fixed on a metal tree-shaped structure and introduced into the chamber where the high-power ultrasound was applied in gas phase. The chamber was based on a cylindrical radiator (310 mm height, 100 mm internal diameter, 10 mm thickness) driven by a piezoelectric transducer (21.8 kHz). An ultrasonic signal was generated and adjusted to minimize the phase between electrical voltage and current by an APG-AC01 dynamic resonance controller (Pusonics, Spain), and the power capability was maintained by an RMX 4050HD power amplifier (QSC, Costa Mesa, CA, USA). An impedance-matching unit APG-AC01 (Pusonics, Spain) was used to optimize the ultrasonic application by electronic methods. This system was located inside an industrial vertical refrigerator ACRV-125-2 (Coreco, Spain) where the temperature was maintained at 17.5 ± 0.3 °C.

The power at which the system was set was 70 W, which meant a power density of 28.8 kW/m^3^. HPU was applied for 120 min (P120), and the results were compared with a sample to which no ultrasound was applied (P0).

The moisture content was measured on samples before and after the ultrasound application and after gastric and intestinal digestion by using the AOAC 934.06 method [21].

### 2.3. Simulated In Vitro Gastrointestinal Digestion

Red bell pepper samples were digested following the in vitro gastric digestion method reported by INFOGEST 2.0 for a static method [22]. So, simulated salivary fluid was prepared with the final concentration of salts in the digestion proposed by this method as follows: 15.1 mM KCl, 3.7 mM KH_2_PO_4_, 13.6 mM NaHCO_3_, 0.15 mM MgCl_2_·6H_2_O, 0.06 mM (NH_4_)_2_CO_3_, 1.1 mM HCl 6M, 1.5 mM CaCl_2_·2H_2_O, and 75 U/mL alpha-amylase; the pH was adjusted to 7.0 with HCl 1M. Simulated stomach fluid: 6.9 mM KCl, 0.9 mM KH_2_PO_4_, 25 mM NaHCO_3_, 47.2 mM NaCl, 0.12 mM MgCl_2_·6H_2_O, 0.5 mM (NH_4_)_2_CO_3_, 15.6 M HCl 6M, 0.15 mM CaCl_2_·2H_2_O, and 2000 U/mL of pepsin from porcine pancreas; the pH was adjusted to 3 with NaOH 1M. Simulated intestinal fluid: 6.8 mM KCl, 0.8 mM KH_2_PO_4_, 85 mM NaHCO_3_, 38.4 mM NaCl, 0.33 mM MgCl_2_·6H_2_O, 0.04 mM CaCl_2_·2H_2_O, 15.6 M HCl 6M, 100 U/mL pancreatin from porcine pancreas and 10 mM bile bovine; the pH was adjusted to 7 with HCl 1M.

After the HPU treatment, the discs of red bell pepper were cut into four equal portions and ∼15 g were mixed with 15 mL of simulated saliva (preheated to 37 °C) for 2 min at 37 °C with orbital agitation (150 U/min) (Rotabit, J.P. Selecta, Barcelona, Spain). After oral digestion, 30 mL of simulated gastric juice were added to the above sample bolus, the pH was adjusted to 3 with HCl 1M and the mixture was incubated for 2 h at 37 °C with orbital agitation (150 U/min). After gastric digestion, the pH was adjusted to 7 with NaOH 1M to stop enzyme activity. Then, 15 mL of simulated intestinal juice were added to 15 g of gastric bolus and the mixture was incubated for 2 h at 37 °C with orbital agitation (150 U/min). All digestion experiments were performed at least in triplicate.

### 2.4. Release of Bioactive Compounds: Total Phenolic Content (TPC) and Antioxidant Activity (AA)

Ethanolic extracts were prepared to determine the antioxidant activity (AA) and the total phenolic compounds (TPC) in the solid samples before digestion (P), after gastric digestion (G), and after gastrointestinal digestion (I). For this, ~1.0 g of sample was weighed, 20 mL of ethanol 85% (*v*/*v*) were added, and the mixture was homogenized with an Ultra-Turrax T25 Digital (Ika, Staufen, Germany) for 60 s, at 13,000 rpm and 4 °C protected from light. The homogenate was refrigerated for 24 h at −20 °C. Afterward, it was centrifuged at 4000 rpm (1252× *g*) for 5 min (ALC 4218, Thermo Scientific, Vantaa, Finland. The supernatant was filtered through Whatman No. 4 paper and stored at −20 °C in the dark until analysis.

The remaining liquids after the gastric and the Intestinal phases (J) were centrifuged at 4000 rpm (1252× *g*) for 10 min. The supernatant was filtered through a 0.45 μm polytetrafluoroethylene (PTFE) filter and stored at 4 °C until analysis. For the liquid from the intestinal phase, the pH was lowered to 2 with 1 M HCl, and absolute EtOH was added at a 1:1 ratio to remove proteins that could interfere with future analyses. Then it was centrifuged, filtered, and stored following the same method described above. These processes were also carried out on the initial juice, to which no red bell pepper sample had been added, to eliminate possible interferences from the used enzymes.

Antioxidant activity (AA) was determined using the FRAP (ferric reducing antioxidant power assay), CUPRAC (cupric reducing antioxidant capacity), and ABTS (radical cation scavenging activity) assays according to Gonzalez-Centeno et al. [23]. The total phenolic content (TPC) was spectrophotometrically determined, from both the ethanolic extracts and the liquids after the gastric and the intestinal phases, according to the method of quantification of Folin–Ciocalteu phenols described by Eim et al. [24]. Absorbance measurements were carried out in a UV/Vis/NIR spectrophotometer Multiskan Spectrum (Thermo Scientific, Finland) at 25 °C, and at a wavelength of 593 (FRAP), 450 (CUPRAC), 734 nm (ABTS), and 745 nm (TPC). Standard curves (0–400 ppm Trolox for AA and 0–300 ppm gallic acid for TPC) were used to correlate absorbance measurements. The experiments were performed in triplicate and the results were expressed for AA as mg of Trolox per 1 g of red bell pepper (on a dry matter basis, dm), and for TPC as mg of gallic acid equivalent (GAE) per 1 g of red bell pepper (on a dry matter basis, dm).

The percentage of release (%) of AA and TPC from red bell pepper has been defined as indicated in Equation (1).
(1)$$\mathrm{Release}\;\left(\%\right)=\frac{C_{0}-C_{i}}{\begin{array}{c} C_{0} \end{array}}\times 100$$
where C_0_ is the initial concentration in the red bell pepper sample before gastrointestinal digestion and C_i_ is the concentration in the sample after gastrointestinal digestion.

### 2.5. Scanning Electron Microscopy (SEM)

The effect of ultrasound on the microstructure of the red bell pepper, before gastrointestinal digestion, was observed in a scanning electron microscope (S-3400N, Hitachi, Japan) at 5 μm resolution, working pressure of 40 Pa, and a voltage of acceleration of 15 kV. Samples were previously freeze-dried for 72 h at −50 °C and 0.3 mbar (LyoQuest, Telstar, Barcelona, Spain). Images were acquired of each sample at 50× magnification.

### 2.6. Optical Microscopy and Image Analysis

According to the methodology described by Vallespir et al. [9], red bell pepper samples were prepared to observe the cell walls of the raw sample, after the HPU treatment, and after both the gastric and the intestinal digestion, by optical microscopy. Samples were fixed in formaldehyde (3.7%), sectioned by using a microtome, and stained with Periodic Acid–Schiff and Hematoxylin Eosin to visualize cell walls [25]. An optical microscope BX41 (Olympus, Tokyo, Japan) and an Optika Camera C-B5+ (CMOS 1/2.5” 5MP–2560 × 1922 pixel) (Optika Microscopes, Bergamo, Italy) were used to obtain microstructure images at 20× magnification. Samples were prepared in duplicate, two sections of each sample were prepared, and at least, twenty optical microscope photographs of each replicate were taken.

An automatic image processing methodology, based on ImageJ 2.0.0 software [26] was used to analyze the optical microscope photographs to measure the area and count the cells in the raw sample and after sonication and after both gastric and intestinal digestion to observe the effect of the process on the microstructure to understand the release of antioxidants and phenolic compounds.

The optical microscope photographs were analyzed following the method described by Reche et al. [27]. First, the contrast of the images was enhanced (“Enhance contrast”), then they were converted to binary (“Make binary”) and the cells were turned to black (“Threshold”). To remove small, misleading particles, a filter was applied to select cells with an area larger than 7000 pixel^2^. The number of cells per unit of tissue surface area was calculated and the percentile profile was calculated using the function “PERCENTIL.EXC” included in Microsoft Excel [28]. The percentile profile represents the percentages of cells whose areas are equal to or less than the obtained value.

### 2.7. Statistical Analysis

Statistical analyses were carried out by R 4.2.2 software [29] together with the rStudio IDE [30]. Parametric one-way analysis of variance (ANOVA) and Tukey’s tests were used to evaluate the existence and extent of significant differences among samples for moisture, AA, TPC, and the number of cells per area unit. Differences at *p* < 0.05 were considered significant.

## 3. Results

### 3.1. Moisture Content

The changes in the moisture content of red bell pepper after the application of 120 min of HPU and after gastric and intestinal digestion are shown in Figure 1a and in Appendix A. The initial moisture content of 11.7 ± 1.1 g H_2_O/g dm in the raw sample was similar to that previously reported by Castro et al. [31] for the same vegetable (12.9 ± 0.4 g H_2_O/g dm). The moisture content decreased by 34 ± 4% after the HPU treatment for 120 min. This result may be because the application of ultrasound has been reported to enhance water transport due to its mechanical energy and mild thermal effect [32]. This effect was previously observed by Martins et al. [33], who applied HPU in the gas phase (50 W) to the drying process of apple peel at 30, 50, and 70 °C, and after the same drying time, the moisture content was lower in samples treated with HPU.

Moisture content was considerably higher (*p* < 0.05) in the samples after gastric digestion, by about 35 ± 5% for the untreated sample (G0) and 17 ± 3% for the sample to which 120 min of ultrasonic treatment was applied (G120), in comparison with the raw sample. On the other hand, the moisture content decreased after intestinal digestion (14 ± 2% and 2.1 ± 0.4% for the untreated (I0) and HPU-treated sample (I120), respectively), compared to the samples after gastric digestion. Hwang et al. [34] performed kale digestion and observed a 16% increase in the moisture content of the samples after gastric digestion. In addition, they also observed a slight decrease of 1% in the moisture content of the samples after intestinal digestion compared to the samples after gastric digestion.

### 3.2. Release of Bioactive Compounds

The antioxidant activity (FRAP, CUPRAC, and ABTS assays) and the content of total phenolic compounds (TPC) were determined on the raw and HPU-treated sample, on these samples after gastric and intestinal digestion; also, in remaining liquids after the gastric and intestinal digestion. The results are also given in Figure 1b–e, respectively and in Appendix A.

In the present study, the following three methods were used to evaluate the AA of both solid red bell pepper and gastric and intestinal phases samples: FRAP, CUPRAC, and ABTS. In this way, a greater and more complete view of the antioxidant compounds present in the red bell pepper was obtained. The average values for the AA of raw red bell pepper were 27.82 ± 0.30 g Trolox/g dm, 31.44 ± 1.85 g Trolox/g dm, and 32.94 ± 1.45 g Trolox/g dm measured by the FRAP, CUPRAC, and ABTS methods, respectively, similar to values previously reported by Rufián-Henares et al. [18] for FRAP (24.2 ± 0.8 g Trolox/g dm) and by Pasli et al. [35] for CUPRAC (24.7 ± 5.9 g Trolox/g dm) and ABTS (21.86 ± 0.07 g Trolox/g dm).

The sample treated for 120 min with HPU showed a significant increase (*p* < 0.05) in the AA measured by FRAP (21 ± 1%) compared to the raw sample. The effect of the use of HPU on AA was also observed by Rodríguez et al. [36], who reported an increase of 45% in the antioxidant activity after drying fresh thyme at 60 °C by applying HPU with a power density of 18.5 kW/m^3^.

An initial TPC value of 14.48 ± 0.30 mg GAE/g dm was obtained for the raw red bell pepper sample, similar to the previously reported value by Cortés-Estrada et al. [37] (13.6 ± 2.0 mg GAE/g dm). 

The sample treated with HPU showed a 37 ± 4% increase in the content of phenolic compounds compared to the raw sample, with significant differences (*p* < 0.05). Do Nascimiento et al. [38] dried passion fruit peel using HPU at an intensity of 154.3 dB and observed that at 40 °C, the TPC increased by 55% compared to samples dried without HPU assistance.

After gastric digestion, the AA of samples significantly (*p* < 0.05) decreased in comparison to the raw sample except for the G0 sample according to the FRAP method (*p* > 0.05). However, the AA decreases were higher in samples treated with HPU (between 22.0 ± 1.7 and 39.9 ± 4.0%) than in those non-treated (between 11.4 ± 1.1 and 28.9 ± 0.5%), in comparison to the same sample before gastric digestion. The TPC of the sample treated with HPU decreased significantly (*p* < 0.05) in comparison to the raw sample, and the decreases were 37.1 ± 2.2% in samples treated with HPU and 6.9 ± 0.3% in those non-treated, in comparison to the same sample before gastric digestion. Furthermore, AA and TPC significantly increased (*p* < 0.05) in the gastric phase of the HPU-treated sample (between 60 ± 7 and 287 ± 12% for AA and 71 ± 8% for TPC) compared to the gastric phase of the untreated sample. 

After intestinal digestion, the AA of samples significantly (*p* < 0.05) decreased in comparison to the raw sample except for the I0 sample according to the ABTS method (*p* > 0.05). However, the AA decreases were higher in samples treated with HPU (between 15.1 ± 1.5 and 54.3 ± 0.8%) than in those non-treated (between 19 ± 3 and 26 ± 4%), in comparison to the same sample before gastrointestinal digestion. No significant differences (*p* < 0.05) were observed in the TPC compared to the raw sample, but the decreases were 36.7 ± 1.6% in samples treated with HPU and 6.9 ± 1.1% in those non-treated in comparison to the same sample before gastrointestinal digestion. The AA showed no significant differences (*p* < 0.05) in the intestinal phase of the HPU-treated sample compared to the phase of the non-treated sample. Nevertheless, TPC showed a significant increase (*p* < 0.05) (55.1 ± 1.2%) in the intestinal phase of the HPU-treated sample compared to the non-treated sample phase. TPC after intestinal digestion decreased in the intestinal phase compared to the TPC of the gastric phase because phenolic compounds are less stable in the intestinal phase [39]. 

No literature has been found about the gastrointestinal digestion of samples to which HPU in the gas phase has been applied. However, Tekin Cakmak [40] observed an increase in the AA of the gastric phase of raspberry samples that had been vacuum dried with the help of an ultrasonic bath at 40 kHz and a power of 590 W. Lafarga et al. [41] studied the gastrointestinal digestion of red bell pepper samples that had been treated using an ultrasonic bath at 4 °C, 40 kHz, and 250 W for 20 min, and observed an increase in the AA in the intestinal phase of the sonicated samples, compared to non-treated samples. Liović et al. [42] applied ultrasound to blueberry puree with a 19.1 mm diameter probe at a frequency of 20 kHz for 3, 6, and 9 min, and observed an increase in the TPC in the HPU-treated samples after both gastric and intestinal phases compared to the untreated sample.

The percentage of release (%) of the bioactive compounds was calculated using the results for the untreated sample and the sample treated with HPU for 120 min. A significant increase (*p* < 0.05) was observed in the release of both AA (between 127 ± 11 and 313 ± 27%) and TPC (755 ± 80%). Ramírez-Moreno et al. [43] treated blueberry samples with ultrasound in liquid phase with a probe at 1500 W, 20 kHz, for 15 and 25 min, performed gastrointestinal digestion, and observed a 21% increase in the release of antioxidant compounds in the HPU-treated samples.

Thus, the application of high-power ultrasound in the gas phase could be an interesting treatment before processes in which high extraction of antioxidants and phenolic compounds is desired. However, used before treatments that promote the degradation of these compounds could contribute to increase degradation, and therefore, not be advisable.

### 3.3. Microstructural Changes Observed by SEM

SEM was used to observe the effect of the treatment on the surface morphology of the red bell pepper before and after the HPU treatment in the gas phase. The SEM images of both the raw and the treated samples can be seen in Figure 2. 

The raw red bell pepper showed a smooth surface with some folds. The part of the red bell pepper observed corresponds to the endocarp; for this reason, this type of morphology is observed and not the polyhedral cells observed by other authors such as Ge et al. [44]. On the other hand, the red bell pepper treated with HPU for 120 min showed a damaged surface, with the appearance of disruptions due to the rupture of the vegetal tissue. Therefore, the application of high-power ultrasound could have a significant effect on the breakdown of plant tissue and, therefore, result in a structure that could release bioactive compounds more easily. Rodriguez et al. [45] dried apples at different temperatures (30, 50, and 70 °C) by applying HPU in the gas phase at two different power densities (between 18.5 ± 0.9 kW/m^3^ and 30.8 ± 0.9 kW/m^3^); observed that the application of HPU caused cell disruptions and the pores became larger than in the raw sample.

### 3.4. Optical Microscope Observation and Image Analysis 

Figure 3 shows the following representative optical microscope images of the red bell pepper samples: raw (P0), treated with HPU (P120), and these samples after in vitro gastric (G) and intestinal digestion (I). Using the method described above, optical microscope photographs were analyzed, and the cell area percentile profiles and the number of cells per unit of tissue surface area were estimated for each sample. The obtained percentile profiles can be seen in Figure 4. The percentiles mean the percentages of cells whose areas are equal to or smaller than the obtained value and can help to evaluate changes in the microstructure by reflecting and quantifying the change in cell size after HPU treatment and after in vitro gastric and intestinal digestion. In addition, to confirm the results observed in the percentile profiles, the results of the number of cells per area are presented in Figure 5.

As can be seen in Figure 4, different percentile profiles were obtained for the different samples. The HPU-treated sample profile was shifted to the left compared to the profile for the raw red bell pepper sample, which means a reduction in cell size. For example, 80% of cells were smaller than 1.55 × 10^−2^ mm^2^ in the raw sample but smaller than 1.11 × 10^−2^ mm^2^ after HPU treatment (ca 29% smaller). This change is in agreement with the results of the number of cells per area (Figure 5), where an increase could be observed of ca 28% in the HPU-treated sample compared to the raw sample, and also, smaller cells can be observed in Figure 3b) highlighted by a black ellipse. As it was mentioned before, the application of ultrasound promoted the loss of water from the sample, which could cause the shrinkage of the cells [46]. The reduction of cell size due to the application of HPU was also observed qualitatively by Vallespir et al. [32], who used HPU in the gas phase with a power density of 20.5 kW/m^3^ to improve the mushroom drying process at 5, 10, and 15 °C.

Samples after in vitro gastric digestion showed larger cell sizes compared to their respective samples before digestion, which can be observed in Figure 3c) highlighted by a black ellipse. For example, 80% of cells were smaller than 1.99 × 10^−2^ mm^2^ in the G0 sample and smaller than 1.39 × 10^−2^ mm^2^ in the G120 sample (28% and 25% larger, respectively). However, although a slight decrease in the number of cells per area can be observed compared to the samples before digestion (22% for the untreated sample and 19% for the treated sample), the differences were not significant (*p* < 0.05). Therefore, in vitro gastric digestion caused alteration of the cell wall structure, which could be attributed to hydration and swelling of the cells. Dalmau et al. [17] observed the changes in the microstructure of dried beetroot using different drying methods after 180 min of in vitro gastric digestion with lower cell numbers per area unit (41–49% less) in comparison with the samples before in vitro digestion.

Samples after gastrointestinal digestion were larger than samples before digestion but smaller than samples after gastric digestion. For example, 80% of cells after gastrointestinal digestion were smaller than 1.67 × 10^−2^ mm^2^ in the I0 sample and smaller than 1.32 × 10^−2^ mm^2^ in the I120 sample. However, although there was a decrease in the number of cells per unit area compared to the undigested samples, and an increase compared to the samples after gastric digestion, the differences were not significant (*p* > 0.05). 

In addition, the effect of the HPU was maintained after both gastric and intestinal digestion because the HPU-treated samples showed smaller sizes and higher numbers of cells per area in all cases. No literature has been found related to image analysis results of samples treated with HPU in the gas phase and after in vitro gastrointestinal digestion. This section may be divided by subheadings. It should provide a concise and precise description of the experimental results, their interpretation, as well as the experimental conclusions that can be drawn.

## 4. Conclusions

The effect of using high-power ultrasound (HPU) as a treatment of red bell pepper on the release of antioxidants and phenolic compounds during in vitro gastrointestinal digestion has been studied. According to the results, it could be concluded that the content of antioxidant and phenolic compounds was affected by the treatment with HPU; thus, both antioxidant activity and phenolic compounds content were higher in the treated sample in comparison to the non-treated sample. Furthermore, the release of these compounds during in vitro gastrointestinal digestion was also higher in the HPU-treated sample. These results could be related to the observed microstructural changes in the vegetable tissue by microscopy combined with image analysis. The HPU application promoted both cell wall disruption and cell shrinkage; thus, a higher number of cells per area unit was observed in the treated samples. Moreover, cell size increased after in vitro gastrointestinal digestion in both samples but to a lesser extent in the sample treated with HPU before digestion.

## Figures and Tables

**Figure 1 antioxidants-12-00356-f001:**
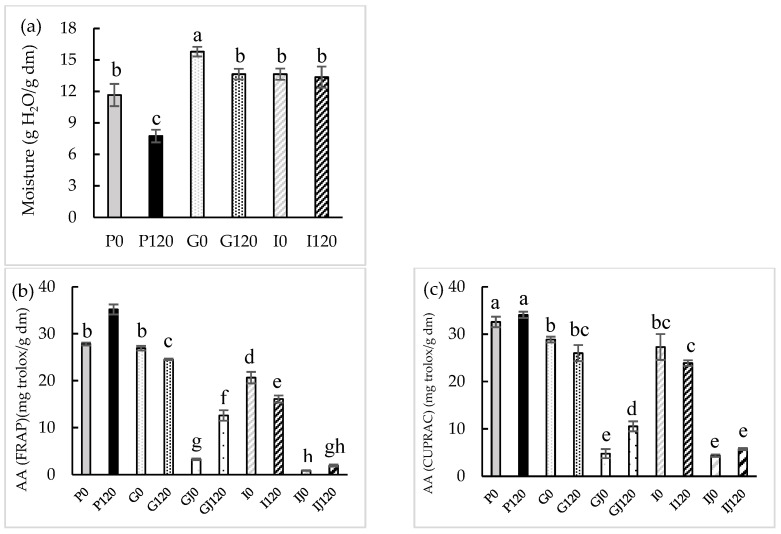
Moisture (**a**), antioxidant activity (AA) (FRAP (**b**), CUPRAC (**c**), and ABTS (**d**) assays) (mg Trolox/g dry matter) and total polyphenol contents (TPC) (**e**) before and after in vitro gastric (G) and intestinal (I) digestions in both solid and juice (J). Different letters for the same parameter indicate significant differences (*p* < 0.05) among samples.

**Figure 2 antioxidants-12-00356-f002:**
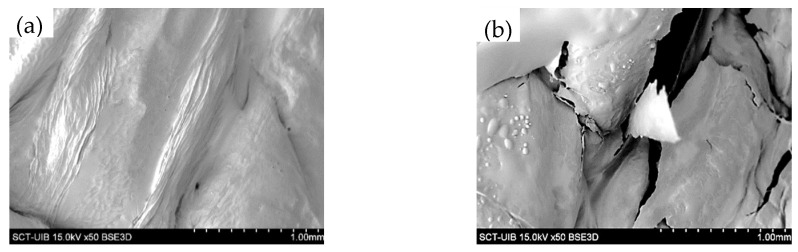
SEM images of red bell pepper samples: raw (**a**) and after 120 min of HPU treatment (**b**).

**Figure 3 antioxidants-12-00356-f003:**
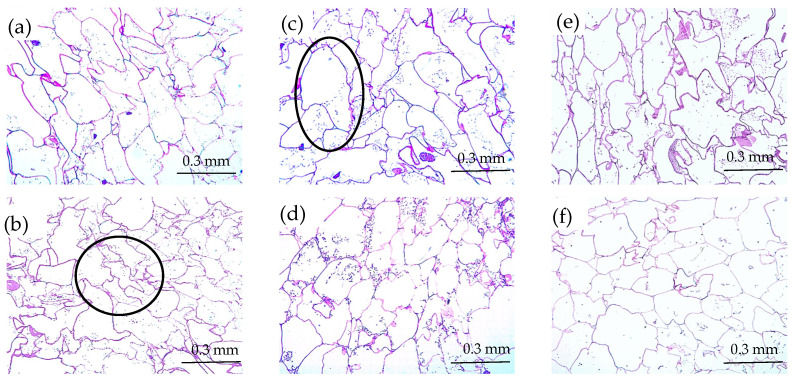
Optical microscope images of the red bell pepper samples: raw (**a**), treated 120 min with HPU (**b**), untreated sample (**c**), and 120 m min treated sample (**d**) after in vitro gastric digestion and untreated sample (**e**) and 120 m min treated sample (**f**) after in vitro intestinal digestion. Black ellipses indicate changes in cell size.

**Figure 4 antioxidants-12-00356-f004:**
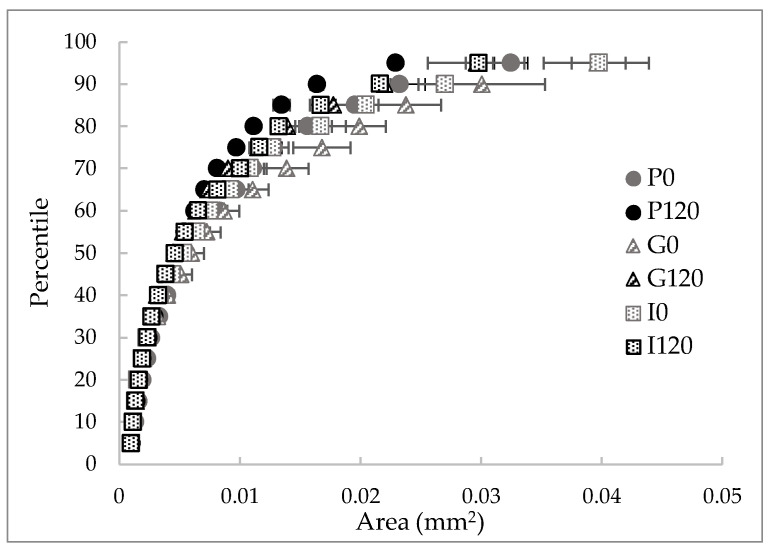
Cell area percentile profiles (percentage of cells that are equal to or smaller than the corresponding area) of the red bell pepper samples: raw (P0), treated 120 min with HPU (P120), and these samples after in vitro gastric (G) and intestinal (I) digestions.

**Figure 5 antioxidants-12-00356-f005:**
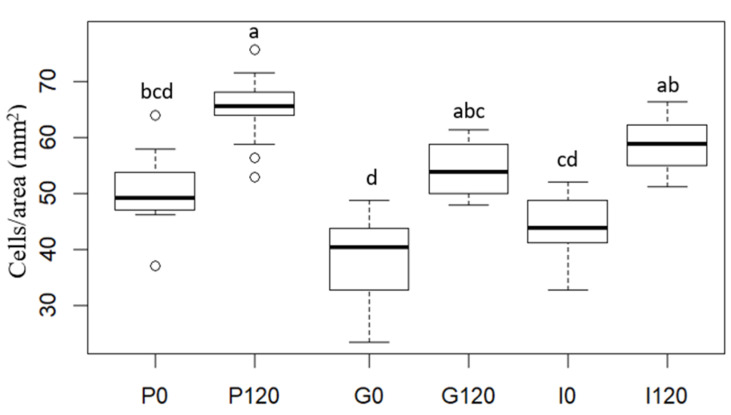
Cell number per unit of tissue surface (cells/area (mm^2^)) of red bell pepper samples: raw (P0), treated 120 min with HPU (P120), and these samples after in vitro gastric (G) and intestinal (I) digestions. Different letters for the same parameter indicate significant differences (*p* < 0.05) among samples.

## Data Availability

Data is contained within the article and Appendix A.

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
