# Peer review of "High-Power Ultrasound in Gas Phase: Effects on the Bioactive Compounds Release from Red Bell Pepper during In Vitro Gastrointestinal Digestion"

_antioxidants, 2023, doi:10.3390/antiox12020356_

Round 1

Reviewer 1 Report

This research paper, “High-power ultrasound in gas phase: effects on the bioactive compounds release from red bell pepper during in vitro gastro intestinal digestion” is interesting and much worthy of investigation. The overall paper reports some interesting results. But, I have some concerns regarding this manuscript.

Comments:

·        P3, S102: ‘HPU was applied for 120 min” why was exposure time kept as 120 min, and its groups not made depending upon a time more or less?

·        P4, S112, S114, and S118: quote the references for composition of simulated salivary fluid, simulated stomach fluid and simulated intestinal fluid.

·        P4, S154: “and 745 (TPC)” also mention the unit of wavelength, either its nm or µm?

·        P4, S161: demonstration of the equation is not clear. It creates confusion with Ci, P0, C0, and P120 so explain it comprehensively.

·        P8, Label images and indicate cell changes through a circle, arrow, or any other way.

·        There are also some nonspecific comments regarding the manuscript that the author should satisfy.

·        Section 2.3, “Simulated in vitro gastrointestinal digestion,” could be written more precisely to avoid confusion.

·        The significance and future aspects of the study are not well described throughout the manuscript and should be added to make the manuscript more meaningful.

·        Limitations of the study must also be added at the end of the discussion section without making a subheading.

·        Send the real-time values of results.

·        Figure 4 is not self-explanatory upload it in a separate system along with values.

·        The conclusion should be short and precise, consisting of basic objectives and outcomes of the study.

Reviewer 2 Report

The authors investigated the effect of pre-Ultrasound in gas phase on the strucuture and antioxidant activity of red pepper. the idea is novel.

Here are a few suggestions: 1. Ultrasound in gas phase is not a common method used, please add a digram to better exaplain the processing; 2. the multple figures need to have letter marks like A, B, C,.. 3. pratical application of this kind of processing should be discussed.

Round 2

Reviewer 1 Report

Thank you for your sincere effort. I have no further comments. The paper can be accepted for publication.